# A Randomized Clinical Study of a Curcumin and Melatonin Toothpaste Against Periodontal Bacteria

**DOI:** 10.3390/biomedicines12112499

**Published:** 2024-10-31

**Authors:** Riccardo Pulcini, Antonio Maria Chiarelli, Bruna Sinjari, Jessica Elisabetta Esposito, Francesco Avolio, Riccardo Martinotti, Vittorio Pignatelli, Luca Pignatelli, Laura Berlincioni, Stefano Martinotti, Elena Toniato

**Affiliations:** 1Department of Innovative Technology in Medicine and Dentistry, Center of Advanced Studies and Technology, University of Chieti, 66100 Chieti, Italy; riccardo.pulcini@unich.it (R.P.); b.sinjari@unich.it (B.S.); j.elisabetta.esposito@gmail.com (J.E.E.); avolio.francesco@gmail.com (F.A.); 2Institute for Advanced Biomedical Technologies (ITAB), Department of Neurosciences, Imaging and Clinical Sciences, University “G. d’Annunzio” of Chieti-Pescara, 66100 Chieti, Italy; antonio.chiarelli@unich.it; 3Recidency Program in Clinical Oncology, Umberto I, University Hospital, La Sapienza, 00142 Rome, Italy; riccardo.martinotti@uniroma1.it; 4Independent Researcher, 66100 Chieti, Italy; pignatelli.vittorio@gmail.com (V.P.);; 5Unit of Clinical Pathology and Microbiology, Department of Medicine, Miulli General Hospital, LUM University, 70021 Acquaviva delle Fonti, Italy; martinotti@lum.it

**Keywords:** curcumin, melatonin, oral health, inflammation modulation, toothpaste

## Abstract

**Background:** The mouth and the oropharyngeal system are home to numerous bacterial species that constitute the so-called oral microbiome and play an important role for the integrity of the oral cavity, influencing the overall health of the body, as demonstrated by several studies. The aim of this study was to evaluate the bacterial modulation potential of a toothpaste (bioredoxin) containing curcumin and melatonin. Both substances have anti-inflammatory properties, as documented in several scientific reports. **Methods**: The in vivo study we present was a single-center, double-blind trial and was conducted in parallel groups. We enlisted 20 volunteers who were randomly assigned to four distinct groups using blinded four different toothpaste preparations: a standard toothpaste indicated as placebo, a toothpaste with curcumin, a toothpaste with melatonin, and a toothpaste with melatonin and curcumin. **Results:** The samples from the gingival tasks were taken at time 0 and after 8 weeks of toothpaste treatment. By evaluating the DNA content of the most significant periodontal bacteria related to the total bacteria count using quantitative PCR assays, including the saprophyte component of the microbiome, we demonstrated that the Curcumin and Melatonin treatment has a statistically relevant effect on decreasing the level of periodontal pathogenic bacteria DNA. The toothpaste with the addition of curcumin and melatonin showed a modulation between t0 and t1 of the *Campylobacter rectus* (14,568 vs. 3532.8) and *Peptostreptococcus micro* (1320.8 vs. 319) bacteria. In addition, a modulation of pathogenic bacteria and saprophytic bacteria was shown. The synergistic action of the two additives would therefore appear to lead to promising results. **Conclusions:** Despite the fact that additional studies may be necessary in evaluating the effect of the Curcumin/melatonin combination in modulating a proposed therapeutic effect on infections of the oropharyngeal apparatus, in this report, we show for the first time that a combination of curcumin and melatonin supplemented using an oral cosmetic vehicle has the capacity to decrease the level of periodontal pathogenic bacteria, possibly ameliorating health and the physiological conditions in the buccal scenario.

## 1. Introduction

The mouth is home to about 700 different species of bacteria, which colonize the entire oral environment, from hard tissues to soft tissues. The acquisition of the oral microbiome evolves steadily throughout life and diversifies over the course of life [1].

The microbiome is the sum of microbes, their genetic information, and the environment in which they interact. It contributes to immunological, physiological, and metabolic functions. In this dynamic, saliva plays a very important role at the microbiological level as it has the function of keeping the microbiome balanced thanks to its antibacterial, fungicidal, and antiviral functions [2,3].

If there is an alteration in this balance, we will find ourselves in the presence of dysbiosis that leads to the uncontrolled proliferation of pathogens that can lead to the manifestations of some pathologies [4].

The oral cavity can be affected by several diseases with high prevalence among human populations, including periodontitis [5,6] and dental caries [5,6,7], which are related to alterations of the oral microbiome and provide evidence on how the composition of the microbiome is related to the state of the disease. In fact, the species of the “red complex” (*Porphyromonas gingivalis*, *Treponema denticola*, and *Tannerella forsythia*) have historically been seen as the main infectious organisms implicated in periodontitis, although many other organisms associated with this disease are involved [8]. Alterations in the oral microbiome can lead, according to some studies, to oral cancer [9] and esophageal cancer [10].

The microbiome of the oral cavity appears to have connections with the microbiomes of the human body, and therefore, in addition to being involved in diseases of the oral cavity, is implicated in a series of systemic diseases. Numerous studies have linked *Fusobacterium nucleatum* with colorectal cancer (CRC) [11] as it is a highly invasive and adherent oral commensal species [12].

Another study showed an abundance of *Porphyromonas gingivalis* and *Aggregatibacter actinomycetemcomitans* in pancreatic cancer samples, both of which are fundamental pathogens in periodontitis [13]. It has been shown that streptococcal species stimulate the production of virulence factors of *Pseudomonas aeruginosa* in a pulmonary environment with cystic fibrosis [14]. Increased systemic inflammation is a major cause of the link with cardiovascular disease [15]. Rheumatoid arthritis has often been linked to periodontitis and involving *Porphyromonas gingivalis* [16].

Neurological disorders such as Alzheimer’s disease have also been associated with the oral microbiome; in fact, the bacterium *Porphyromonas gingivalis* has been identified in the brain, which, in mice, has increased amyloid plaques, leading to the discovery of how gingipains proteases are neurotoxic and favor the inhibition of the tau protein [17,18,19].

Endocrine system disorders, such as diabetes, seem to be associated with dysbiosis of the oral microbiome; in fact, in one study, it was seen that the prevalence of periodontitis was 60% in subjects with diabetes and 36% in subjects without diabetes [20,21].

The importance of the microbiome is increasingly evident both for the health of the oral cavity and for human health; therefore, the understanding and possibility of possible modulation of the same are fundamental. Even more important is to study how compounds of natural origin, such as curcumin and melatonin, can play a decisive role.

Curcumin is a polyphenolic substance present in Turmeric, a plant that is part of the Turmeric group, which in turn is part of the botanical family of Zingiberacea. It is a yellow pigment extracted from the roots of turmeric plants. Curcumin has been used for centuries as a natural remedy for many diseases, especially in Ayurvedic medicine [22]. The scientific literature is very rich in studies on the properties of curcumin and its possible applications in the medical, dental, and pharmacological fields [23,24,25,26,27,28]. This polyphenol has been shown to have anti-inflammatory, hypoglycemic, antioxidant, healing, and antimicrobial activities [29,30]. Various studies show that curcumin has anti-inflammatory properties due to its ability to inhibit enzymes such as cyclooxygenase, lipoxygenase, and nitric oxide synthetase, which are key molecules of inflammatory reactions. Curcumin, therefore, lends itself to being used for the treatment of various inflammatory diseases [31,32], having a powerful anti-inflammatory power and possessing antibacterial activity against periodontal bacteria [33].

Curcumin has been shown to exhibit anti-inflammatory biological activity [34] in gingival fibroblasts, which, when stimulated by lipopolysaccharide (LPS), can activate the nuclear factor kappa-B (NF-κB) signaling pathway and produce inflammatory cytokines such as IL-1β and TNF-α. Extensive research has shown that transcription factor NF-κB is a key component of the inflammatory process [35,36].

Melatonin (N-acetyl-5-methoxytryptamine) is synthesized and secreted by the pineal gland and other organs. Melatonin secretion is controlled by an endogenous circadian timing system and transmits light–dark cycle information to the body, thus organizing its seasonal and circadian rhythms. Melatonin has powerful antioxidant effects, plays an immunomodulatory role, and promotes bone formation; these effects can contribute to the protection of the oral cavity and could contribute to the regeneration of alveolar bone through the production of type I collagen and through the modulation of osteoblastic and osteocyte activity [37]. Several studies have been interested in demonstrating how melatonin can be a promising therapeutic application for anti-inflammatory treatment, thanks to its characteristics. In animals, it has been observed that the levels of pro-inflammatory cytokines, including TNF-α, IL-1β, and INF-γ, are elevated; these alterations have been attenuated by melatonin. Both local and systemic anti-inflammatory cytokine IL-10 levels are markedly elevated by LPS and were further increased when melatonin was administered [38,39,40].

Melatonin is also released into the saliva by the acinar cells of the major salivary glands and through gum fluid. It is likely that the functions of melatonin in the oral cavity are mainly related to its anti-inflammatory and antioxidant activities. These actions could reduce inflammation of the gum and periodontium [41].

Melatonin, therefore, is considered a powerful molecule in the management of a wide variety of diseases with inflammatory etiology [42,43].

Given the properties of melatonin and curcumin, it is possible to hypothesize how the two polyphenols can have a synergistic effect in bacterial modulations and modulations in inflammation of the oral cavity.

The aim of this work was to understand the effect of the use of a curcumin and melatonin toothpaste (Vtx-bioredoxin^®^) on the bacterial species considered (*Aggregatibacter actinomycetemcomitans*, *Porphyromonas gingivalis*, *Tannerella forsythia*, *Treponema denticola*, *Fusobacterium nucleatum*, *Prevotella intermedia*, *Peptostreptococcus micro*, *Campylobacter rectus*, *Eikenella corrodens*) at a qualitative and quantitative level through RT-PCR and on the modulation of the periodontal indices taken into consideration, such as the Full-Mouth Plaque Score (FMPS) and Full-Mouth Bleeding Score (FMBS). For more information, a toothpaste containing an addition of only curcumin, a toothpaste containing an addition of only melatonin, and a standard toothpaste were also evaluated. The changes observed in the sample under consideration suggest how a toothpaste with polyphenolic substances with anti-inflammatory and antibacterial effects, widely studied in the literature, can provide benefits for the health of the oral cavity.

## 2. Materials and Methods

This study was classified as non-pharmacological clinical trial n.2200 on 7 October 2020, was approved by the Ethics Committee of the Provinces of Chieti and Pescara on 6 April 2021, and was performed in accordance with the articles of the Helsinki Declaration. All volunteers read and signed the informed consent given.

### 2.1. Volunteers Description

In this study, volunteers with a healthy and robust constitution were selected. The inclusion criteria were subjects aged between 18 and 55 years, of both sexes, who did not have chronic diseases and did not take antibiotics or anti-inflammatories, who did not have diseases affecting the oral cavity (caries, gingivitis, and periodontitis), or who were not pregnant.

### 2.2. Study Design

This was a monocenter, randomized, double-blind, parallel-group study comparing a toothpaste containing curcumin, a toothpaste with melatonin, a toothpaste with melatonin and curcumin (Vtx-bioredoxin^®^), and a placebo group.

Volunteers were randomized by the Vitalex H.C. using the randomized function of Excel (Microsoft Office version 10.0). Placebo and tested toothpastes did not have any difference with regard to smell and test, as also certified from the factory. All the researchers involved in this project and the enrolled patients did not know which of the four compounds were used.

Below is a flowchart representing the phases of the study performed (Figure 1). Twenty volunteers (Table 1) were recruited following the inclusion and exclusion criteria, to whom a bacterial plaque sample was carried out and periodontal indices (FMPS and FMBS) were evaluated; moreover, instructions were given regarding home hygiene to be performed during the treatment period. After 60 days of treatment with the various toothpastes randomly assigned to the volunteers, a new sample of bacterial plaque was collected and a new evaluation of the periodontal indices was performed.

### 2.3. Description of Plaque Sampling

The sample was taken using sterile paper cones, which were inserted into the gingival grooves of the volunteers for 1 min and then inserted into sterile tubes for the analysis of periodontal bacteria [44].

### 2.4. DNA Extraction

To extract the DNA from the sample, the following reagents were inserted into the tube containing the biological material: 180 μL Buffer ATL (Qiagen, Hilden, Germany), 150 μL Buffer PBS 1X (Corning, Somerville, MA, USA), and 20 μL Proteinase K (Qiagen). Subsequently, the sample was placed in a water bath at 56 °C for 10 min with oscillation at 80 rpm. At the end, a cell lysate product was obtained from which it was possible to isolate DNA. An amount of 400 μL of 96% ethanol was added and the sample was transferred to a filter tube. The cellular DNA debris was separated through centrifugations.

### 2.5. DNA Amplification

Once the DNA was purified, the hypervariable region of the bacterial gene 16S was amplified through the Real-Time PCR method. The following reaction mix was then prepared for each sample: 5 μL TaqPathTM ProAmpTM Multiplex Master Mix (ThermoFisher, Waltham, MA, USA), 0.5 μL probe (Ba04930791, ThermoFisher), 3 μL DNA, and 1 μL H20 (Table 2).

The amplification reaction was performed on the Rotor-Gene Q (Qiagen, Hilden, Germany) instrument, using the protocol shown in Table 2. Subsequently, analysis of the test bacteria was performed in Table 3.

A Real-Time PCR multiplex was carried out so that different bacteria could be analyzed simultaneously through the use of different fluorescent probes specific to each bacterium.

A triplex was then prepared with the following volumes: 7.5 μL TaqPath” ProAmp” Multiplex Master Mix (ThermoFisher), 0.7 μL Complex 1, labeled in FAM (ThermoFisher), 0.7 μL Complex 2, labeled in VIC (ThermoFisher), 0.7 μL Complex 3, labeled in JUNE (ThermoFisher), 3 μL DNA, and 2.4 μL H20. Each complex consists of a forward primer, a reverse primer, and a probe conjugated with a fluorophore (FAM, VIC, or JUNE), which pairs with the sequence of the target DNA, allowing its detection.

The amplification reaction was performed on the Rotor-Gene Q (Qiagen) instrument using the protocol shown in Table 2.

### 2.6. Statistical Analysis

The normal distribution of variables was evaluated using the Kolmogorov–Smirnov test. Multiple two-way repeated measures ANOVAs were performed for investigating the effect of study group (between factors: P, C, M, and CM) and time (within factors: T0 and T1). Because of the experimental hypothesis, the focus of the analysis was on the statistical significance of the interaction effect “Group × Time”. The variables with a significant interaction were selected and fed to a post hoc analysis. The post hoc analysis was implemented computing multiple unpaired *t*-tests comparing the longitudinal change (T1–T0) of the selected variables between groups. A *p* < 0.05 was considered statistically significant for both the two-way ANOVAs and the *t*-tests (False Discovery Rate correction was implemented to account for multiple comparisons).

## 3. Results

Twenty-four volunteers after a blood draw were screened for blood count with formula, AST/ALT, GGT, creatinine, albumin, glycemia, urea nitrogen, HDL/LDL, total cholesterol, total and fractionated bilirubin, Na+/K+, ESR, PCR, PT, PTT, fibrinogen, cholinesterase, urine test, HIV antibodies, HCV, hepatitis B virus HbsAg, HbeAg, HbcAg IgM, cannabinoid screening, beta-HCG for female volunteers, prolactin, cortisol, and serotonin. A general visit was carried out to test the good health conditions and ECG monitoring of the subjects. During the screening visit, the various drug intakes (steroidal and non-steroidal anti-inflammatory drugs and antibiotics) were investigated in the volunteers in order not to alter the bacterial results obtained.

Following the screening, twenty volunteers were recruited, the average age of the volunteers enrolled was 30.7 years, of which twelve were males and eight were females (Table 1). Four volunteers withdrew following screening for personal reasons. The home oral hygiene instructions to be followed for the entire duration of the study were illustrated to the 20 remaining volunteers, and a plaque sample was performed as previously described. They were subsequently divided into four groups and, for each group, a toothpaste was distributed (standard toothpaste, toothpaste with added curcumin, toothpaste with added melatonin, toothpaste with added melatonin and curcumin). After 60 days, the volunteers reconvened, and new plaque was taken (the flow of participants is described in Figure 1).

In Figure 2A, we observed how in treatments with curcumin, melatonin, and curcumin and melatonin, the percentage of saprophytic bacteria present in the samples taken increased after the treatment. The percentage of saprophytes was obtained by subtracting the pathogenic bacteria from the total bacterial load. In Figure 2B, we observed the percentage difference in the various treatments, showing that the best composition is the toothpaste with the addition of curcumin and melatonin.

In Figure 3A, we observed how the pathogenic bacteria decreased in the samples subjected to treatment, with a greater result in the treatment with a toothpaste with the addition of curcumin and melatonin. Furthermore, in Figure 3B, it is possible to observe the percentage difference between the various pre-treatment and post-treatment applications, showing how the CM treatment reduces the load of the pathogenic periodontal bacteria taken into consideration.

In particular, we reported the results of two bacteria that showed a high response to the various types of treatment: an orange cluster bacterium (CR—*Campylobacter rectus*) (Figure 4) and a red cluster bacterium (PM—*Peptostreptococcus micro*) (Figure 5). In these images, we observed how, in the placebo (standard toothpaste), both bacteria increase their numbers between T0 and T1, while, in the CM treatment (toothpaste with the addition of melatonin and curcumin), there is an improvement with respect to a beneficial decrease in pathogenic bacteria.

## 4. Discussion

The health of the oral cavity is particularly relevant since it can have a key function in systemic disorders. As a consequence, it is crucial to identify agents that can naturally modulate the inflammatory state and the oral microbiome. The healthy properties of melatonin and curcumin, the two compounds that we have used for our investigation, have so far been extensively demonstrated, and studies in the literature have reported their remarkable efficacy.

From our double-blind randomized clinical trial, we obtained some results worthy of note. Our results showed a modulation of some oral bacteria, such as Peptostreptococcus micro (PM) and Campylobacter rectus (CR) due to the administrated treatment, whereas for others, we obtained data without statistical significance. In any case, it should be considered that our investigation has several limitations. The first and most important limit of this study was the low number of the enrolled patients (twenty volunteers). The study was intended to be a preliminary investigation, which accounts for the limited sample size that was enrolled.

The changes in the saprophytic and pathogenic bacteria populations reflect the different treatment that was administered in each group. As shown in Table 4, the group of volunteers who had been administered the standard toothpaste (P) had an increase in the pathogenic bacteria, whereas the load of the saprophytic microorganisms was reduced. In the same group, we also observed a substantial increase in the concentration of Peptostreptococcus micro and Campylobacter rectus that could be attributable to the deterioration of the oral health of the subjects considered, since, as described in the rationale of the study, the toothpastes used in the various groups were analogs, with the exception of the added percentages of Curcumin and Melatonin that it is currently covered by a patent. Also, group C, whose members had been administered toothpaste with the curcumin additive, showed an increase in pathogenic bacteria, whereas the concentrations of CR and PM decreased. In group M, which had been administrated with a toothpaste with the melatonin additive, we observed a decrease in pathogenic bacteria and an increase in saprophytic bacteria. In this group, CR showed an increase in concentration, while the concentration of PM decreased. We observed the most encouraging results in the CM group, which received toothpaste with the additions of curcumin and melatonin, showing a decrease in pathogenic bacteria and an increase in saprophytic bacteria. A significant decrease was also observed in the T1 concentrations of CM and PM. We attributed this drastic improvement to the synergistic action of the two compounds added to the toothpaste.

Although these are promising results, it should be considered that this investigation was designed as a preliminary study. Future studies with a larger sample of patients should be performed, with longer periods of treatment and, possibly, with intermediate phases to collect new samples of bacterial plaque in order to test any possible bacterial growth curves and modulations.

## 5. Conclusions

From this preliminary study, we could observe how the synergistic action of these two products could indeed modulate bacterial populations. Further studies are warranted to broaden the sampling of volunteers in order to better understand the bacteriostatic or bactericidal dynamics of these natural compounds and which bacterial classes respond most effectively to the treatment with a toothpaste with an addition of melatonin and curcumin.

A toothpaste enriched with natural products with demonstrated benefits for antimicrobial and anti-inflammatory properties, such as curcumin and melatonin, could be used for targeted therapies to decrease and/or modulate the microbiotic population of the oral cavity, in which there are about 700 species of bacteria, in order to guarantee oral health.

## Figures and Tables

**Figure 1 biomedicines-12-02499-f001:**
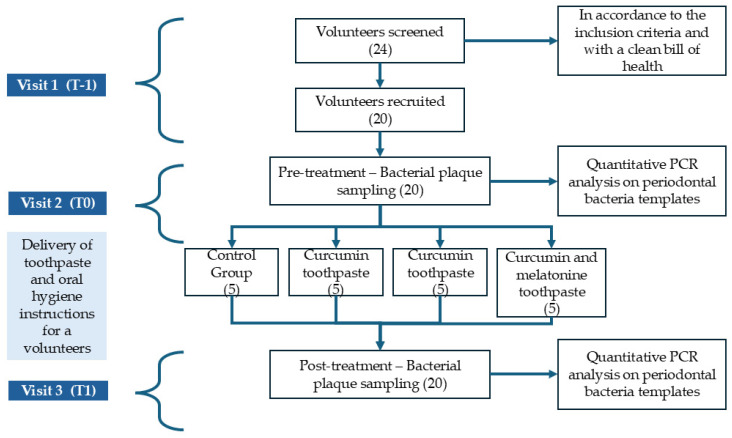
Flowchart representing the study design.

**Figure 2 biomedicines-12-02499-f002:**
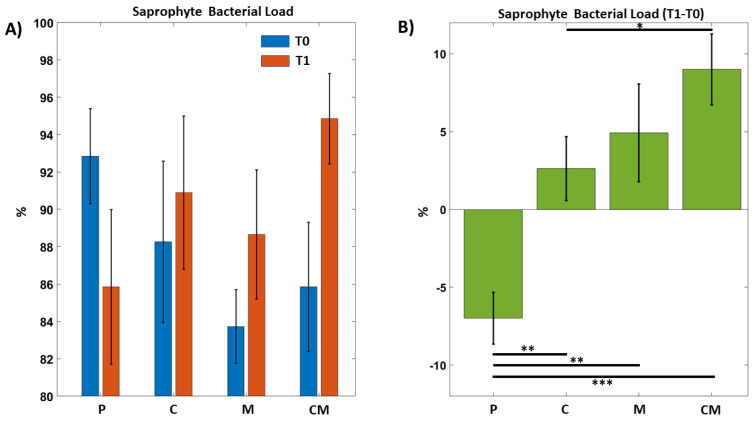
Saprophyte bacterial load on different groups before and after treatments. Two-way repeated Anova, Interaction Effect Group by Time: *p* = 1.2 × 10^−4^ Paired (T1–T0) *T*-tests between groups (significant ps). CM vs P: 4.46 × 10^−4^. M vs P: *p* = 0.0096. C vs. P: *p* = 0.0062. CM vs. C: *p* = 0.049. (**A**) The groups are indicated by P = Placebo, C = Curcumin, M = Melatonin, CM = Curcumin and melatonin treatments. T0 (blue) indicates the beginning of treatments; T1 (orange) stands for 8 weeks of treatments. (**B**) Various percentages calculated as a t0 ratio on different groups before and after treatments. * *p* < 0.05, ** *p* < 0.01, *** *p* < 0.001.

**Figure 3 biomedicines-12-02499-f003:**
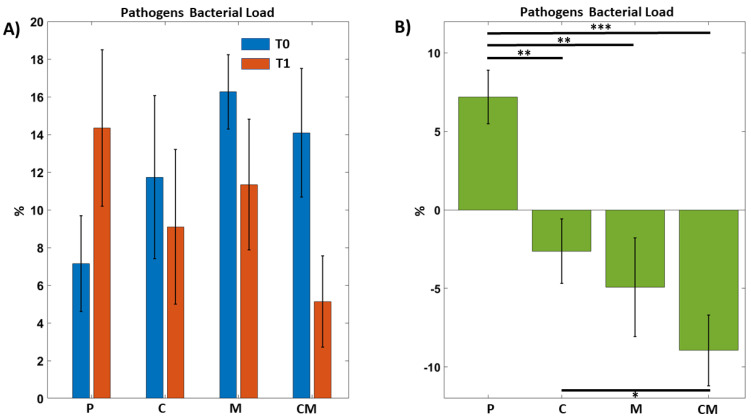
Pathogen bacterial load on different groups before and after treatments. Two-way repeated Anova, Interaction Effect Group by Time: *p* = 1.17 × 10^−4^ Paired (T1–T0) *T*-tests between groups (significant ps) CM vs. P: 4.72 × 10^−4^. M vs.P: *p* = 0.009. C vs. P: *p* = 0.0065. CM vs. C: *p* = 0.048. (**A**) The groups are indicated by P = Placebo, C = Curcumin, M = Melatonin, CM = Curcumin and melatonin treatments. T0 (blue) indicates at the beginning of treatments, T1 (orange) stands for 8 weeks treatments. (**B**) Various percentages calculated as a t0 ratio on different groups before and after treatments. * *p* < 0.05, ** *p* < 0.01, *** *p* < 0.001.

**Figure 4 biomedicines-12-02499-f004:**
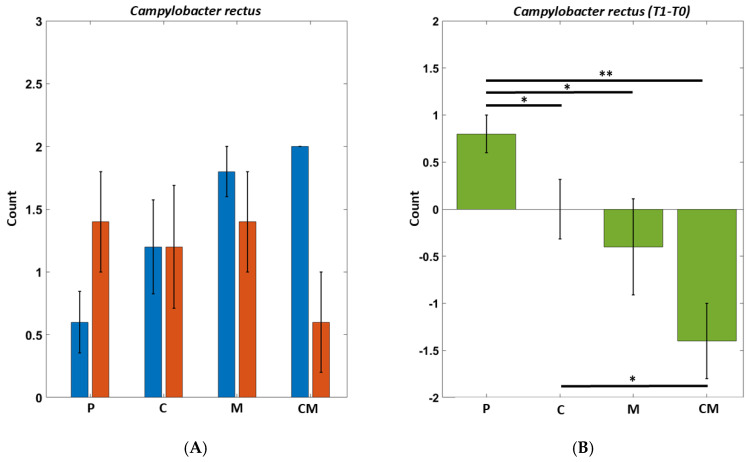
*Campylobacter rectus* on different groups before and after treatments. Two-way repeated Anova, Interaction Effect Group by Time: *p* = 0.0012. (T1–T0) *T*-tests between groups (significant ps). CM vs. P: 0.0012. M vs. P: *p* = 0.048. C vs. P: *p* = 0.049. CM vs. C: *p* = 0.025. (**A**) The groups are indicated by P = Placebo, C = Curcumin, M = Melatonin, CM = Curcumin and melatonin treatments. T0 (blue) indicates the beginning of treatments; T1 (orange) stands for 8 weeks of treatments. (**B**) Various percentages calculated as a t0 ratio on different groups before and after treatments. * *p* < 0.05, ** *p* < 0.01.

**Figure 5 biomedicines-12-02499-f005:**
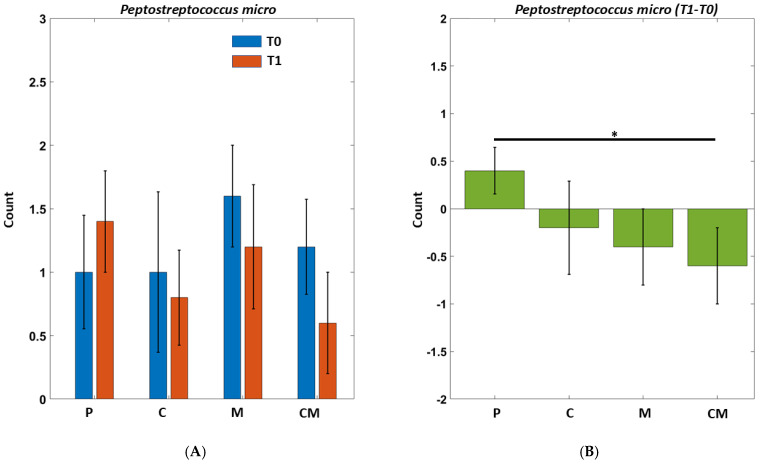
*Peptostreptococcus micro* on different groups before and after treatments. Two-way Repeated Anova, Interaction Effect Group by Time: *p* = 0.0034. Paired (T1–T0) *T*-tests between groups (significant ps). CM vs. P: *p* = 0.047. (**A**) The groups are indicated by P = Placebo, C = Curcumin, M = Melatonin, CM = Curcumin and melatonin treatments. T0 (blue) indicates the beginning of treatments; T1 (orange) stands for 8 weeks of treatments. (**B**) Various percentages calculated as a t0 ratio on different groups before and after treatments. * *p* < 0.05.

**Table 1 biomedicines-12-02499-t001:** Summary diagram with male/female ratio and age range.

Volunteers	Volunteers Number	Female/Male	Age (Mean/Range)
Screened volunteers	24	10/14	31.13/25–38
Enlisted volunteers	20	8/12	30.7/26–38

**Table 2 biomedicines-12-02499-t002:** Fluorescence acquisition.

Hold	95 °C	10 min
Cycling (40 CYCLES)	95 °C	15 s
	60 °C	60 s

**Table 3 biomedicines-12-02499-t003:** List of periodontal pathogenic bacteria analyzed as DNA templates in quantitative RT-PCR assays. Colors indicate the different pathogenicity from red to green (red is the highest and green is the least relevant).

Bacterial Species	Acronym
*Aggregatibacter actinomycetemcomitans*	AA
*Porphyromonas gingivalis*	PG
*Tannerella forsythia*	TF
*Treponema denticola*	TD
*Fusobacterium nucleatum*	FN
*Prevotella intermedia*	PI
*Peptostreptococcus micro*	PM
*Campylobacter rectus*	CR
*Eikenella corrodens*	EC

**Table 4 biomedicines-12-02499-t004:** PCR quantitative data and percentages of pathogens bacteria and saprophyte bacteria.

TREATMENT	TIMING	CR	PM	PATHOGENS (%)	SAPROPHYTE (%)
**P**	T0	2987.2	3499.2	7.156	92.844
	T1	13434	15484	14.344	85.856
**C**	T0	3383.8	33694	11.736	88.264
	T1	4186	444.8	9.102	90.898
**M**	T0	3460	5842	16.268	83.732
	T1	5403	4130	11.344	88.658
**CM**	T0	14568	1320.8	14.086	85.862
	T1	3532.8	319	5.1398	94.86

## Data Availability

Data are available from the corresponding author upon reasonable request.

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
