# Peer review of "A Randomized Clinical Study of a Curcumin and Melatonin Toothpaste Against Periodontal Bacteria"

_biomedicines, 2024, doi:10.3390/biomedicines12112499_

Round 1

Reviewer 1 Report

Comments and Suggestions for Authors

Dear authors of the article!

The topic of your research is interesting and relevant, as there is a constant search for effective natural components for hygiene and prevention of oral diseases. More and more patients suffer from periodontitis and gingivitis.

However, the design of the study and the description of the results with a number of errors:

1. In the introduction, there is no convincing data from the literature that would indicate the anti-inflammatory and antibacterial effect of curcumin and melatonin. Mistakes were made in the spelling of microbes in Latin (Tannarella forsythia), in some cases the outdated name Tannarella forsythensis is used. In addition, all names of microbes in Latin should be written in italics. The name of the microbe species is always written with a small letter after the genus of the microorganism, there are many such errors in the text, for example, when describing the results and in the title of the drawings.

2. A small number of volunteers. The conclusions are statistically unreliable with so many studies.

3. The number of volunteers does not match in different parts of the article – in the summary and figure 20 people, in the description of materials and methods 12. The main information about the characteristics of the groups should be in the text, not just in Figure 1.

4. The figures do not contain all the necessary signatures to understand the results.

5. There is no effect from the use of pastes on the amount of DNA of the main periodontal pathogens (A.actinomycetemcomitans, Prevotela intermediis, Porphyrominas gingivalis, etc.), respectively, the conclusion about the effectiveness of pastes with the addition of curcumin and melatonin is not credible.

The article needs serious revision.

Comments on the Quality of English Language

Extensive editing of English language required.

Author Response

Dear Editor,

Attached please find our reply to all of the criticisms raised by the reviewer. We feel that we did all our best to take them into consideration, to modify our text, in a way that it may better represent and discuss the main findings of our study. The following notes address in detail each specific point and give evidence of modifications in the text.

Answers to Reviewer 1 Comments:

  1. In the introduction, there is no convincing data from the literature that would indicate the anti-inflammatory and antibacterial effect of curcumin and melatonin. Mistakes were made in the spelling of microbes in Latin (Tannarella forsythia), in some cases the outdated name Tannarella forsythensisis used. In addition, all names of microbes in Latin should be written in italics. The name of the microbe species is always written with a small letter after the genus of the microorganism, there are many such errors in the text, for example, when describing the results and in the title of the drawings.

We agreed with the reviewer. We performed an extensive revision to correct all the microorganism names.

  1. A small number of volunteers. The conclusions are statistically unreliable with so many studies.

We agreed with the reviewer. Accordingly with the sponsor, we performed a preliminary study. The small number of the enrolled patients has been reported in the discussion section as a limitation of the study: “In any case, it should be considered that our investigation has several limitations. The first and most important limit of this study was the low number of the enrolled patients (twenty volunteers). The study was intended to be a preliminary investigation, which accounts for the limited sample size that was enrolled.” We also changed the conclusion.

  1. The number of volunteers does not match in different parts of the article – in the summary and figure 20 people, in the description of materials and methods 12. The main information about the characteristics of the groups should be in the text, not just in Figure 1.

We agreed with the reviewer. We corrected the number of patients throughout the text.

  1. The figures do not contain all the necessary signatures to understand the results.

We thank the reviewer. We have checked the figures. The signatures are descripted into the caption.

  1. There is no effect from the use of pastes on the amount of DNA of the main periodontal pathogens (actinomycetemcomitans, Prevotela intermediis, Porphyrominas gingivalis,etc.), respectively, the conclusion about the effectiveness of pastes with the addition of curcumin and melatonin is not credible.

We thank the reviewer. In our results, we observed a modulation in the principal periodontal pathogens, but we know that we have enrolled a small number of patients. Further studies with a larger group of patients are warranted.

Reviewer 2 Report

Comments and Suggestions for Authors

The present manuscript entitled "A randomized clinical study of a curcumin and melatonin toothpaste against periodontal bacteria" describes the antimicrobial properties of the curcumin and melatonin toothpaste. Though the study looks innovative and well-attempted, I suggest some significant modifications prior to further consideration. My comments and suggestions are given below;

1. The abstract lack quantitative data; though authors have given the overview of finding, there require clear output of major findings

2. The introduction is written under different subheadings. It will be better to convert the writing without any subheadings. It ensures the flow of reading.

3. The citation stlyle in the manuscript text seems to be wrong; it needs to be given in the correct format

4. In the methodology, the first and most important question is the preparation of tooth paste. How the tooth paste was prepared? What was the individual concentrations of curcumin and melatonin in it? How the concentrations were selected?

5. The position of tables legends (Table 1, 2) should be checked

6. The discussion is too short and it failed to address the observed results.

Comments on the Quality of English Language

1. There are plenty of typographic errors and spacing issues. Authors need to thoroughly proof read the article before revision.

Author Response

Dear Editor,

Attached please find our reply to all of the criticisms raised by the reviewer. We feel that we did all our best to take them into consideration, to modify our text, in a way that it may better represent and discuss the main findings of our study. The following notes address in detail each specific point and give evidence of modifications in the text.

Answers Reviewer 2 Comments:

  1. The abstract lack quantitative data; though authors have given the overview of finding, there require clear output of major findings

We agreed with the reviewer. We added the following sentences to the abstract: “The toothpaste with the addition of curcumin and melatonin showed a modulation between t0 and t1 of the Campylobacter rectus (14568 vs 3532.8) and Peptostreptococcus micro (1320.8 vs 319) bacteria. In addition, modulation of pathogenic bacteria and saprophytic bacteria was shown. The synergistic action of the two additives would therefore appear to lead to promising results.”

  1. The introduction is written under different subheadings. It will be better to convert the writing without any subheadings. It ensures the flow of reading.

We agreed with the reviewer. We have removed all the subtitles in the introduction section.

  1. The citation stlyle in the manuscript text seems to be wrong; it needs to be given in the correct format

We agreed with the reviewer. We performed an extensive revision of the citation style throughout the text.

  1. In the methodology, the first and most important question is the preparation of tooth paste. How the tooth paste was prepared? What was the individual concentrations of curcumin and melatonin in it? How the concentrations were selected?

We thank the reviewer. Unfortunately, the formulation of the toothpaste is covered by patent.

  1. The position of tables legends (Table 1, 2) should be checked

We agreed with the reviewer. We changed the position of the table captions.

  1. The discussion is too short and it failed to address the observed results.

We agreed with the reviewer and we decided to implement the discussion section.

There are plenty of typographic errors and spacing issues. Authors need to thoroughly proof read the article before revision.

We agreed with the reviewer. We performed an extensive revision of the article.

Reviewer 3 Report

Comments and Suggestions for Authors

1. The study involves only 20 participants, which is a relatively small sample size for a clinical trial aimed at assessing the efficacy of a new treatment. The small sample size may limit the generalizability of the findings and the statistical power to detect significant differences between groups.

2. The study is described as randomized and double-blind, howver, there is a lack of detail regarding how randomization was carried out and how blinding was maintained. 

3. The authors should provide the ages of each group.

4. The odor of the toothpaste in the placebo group was different from the odor of the toothpaste in the Curcumin group, so it is hard to say that this was a double-blind clinical trial.

5. Commercial toothpaste is commonly available with antibacterial properties, but what is the reason for the increase in the total pathogenic bacteria count in the placebo group after using the toothpaste? If toothpaste that does not have antibacterial properties was used, the reviewer feels that there are ethical concerns.

6. The placebo group at the T0 period had significantly lower pathogens than several other treatment groups, and it looks like this was not a randomized clinical trial.

7. The conclusion states that the use of curcumin and melatonin toothpaste "provides clear benefits" for oral health, which may be an overstatement given the study's limitations, particularly the small sample size and the short duration of the intervention.

8. Detailed PCR data processing and PCR results such as the relative abudance of Aggregatibacter actinomycetem comitans, Porphyromonas gingivalis... should be provided.

Comments on the Quality of English Language

Minor editing of English language required.

Author Response

Dear Editor,

Attached please find our reply to all of the criticisms raised by the reviewer. We feel that we did all our best to take them into consideration, to modify our text, in a way that it may better represent and discuss the main findings of our study. The following notes address in detail each specific point and give evidence of modifications in the text.

  1. The study involves only 20 participants, which is a relatively small sample size for a clinical trial aimed at assessing the efficacy of a new treatment. The small sample size may limit the generalizability of the findings and the statistical power to detect significant differences between groups.

    We agreed with the reviewer. Accordingly with the sponsor, we performed a preliminary study. The small number of the enrolled patients has been reported in the discussion section as a limitation of the study: “In any case, it should be considered that our investigation has several limitations. The first and most important limit of this study was the low number of the enrolled patients (twenty volunteers). The study was intended to be a preliminary investigation, which accounts for the limited sample size that was enrolled.”

    2. The study is described as randomized and double-blind, howver, there is a lack of detail regarding how randomization was carried out and how blinding was maintained. 

    4. The odor of the toothpaste in the placebo group was different from the odor of the toothpaste in the Curcumin group, so it is hard to say that this was a double-blind clinical trial.

    We agreed with the reviewer. We added the following paragraph to the method section: “Volunteers were randomized by the Vitalex H.C. using the randomized function of Excel (Microsoft Office 10.0). Placebo and tested toothpastes did not have any differences with regard to smell and taste, as certified also from the factory. All the researchers involved in this project and all the enrolled patients did not know what of the four compounds were used.”

    5. Commercial toothpaste is commonly available with antibacterial properties, but what is the reason for the increase in the total pathogenic bacteria count in the placebo group after using the toothpaste? If toothpaste that does not have antibacterial properties was used, the reviewer feels that there are ethical concerns.

    We thank the reviewer. The placebo group used a standard toothpaste. The reason why the count and the percentage of pathogen microorganisms after 8 weeks from the beginning of the study is that normally the bacterial load increase over time. It is the same for the other groups, but our research seems to prove that the tested toothpaste is more bacteriostatic. We have discussed that in the discussion section.

    6. The placebo group at the T0 period had significantly lower pathogens than several other treatment groups, and it looks like this was not a randomized clinical trial.

    We thank the reviewer. Volunteers were randomized by the Vitalex H.C. using the randomized function of Excel (Microsoft Office 10.0).

    7. The conclusion states that the use of curcumin and melatonin toothpaste "provides clear benefits" for oral health, which may be an overstatement given the study's limitations, particularly the small sample size and the short duration of the intervention.

    We agreed with the reviewer. We changed the conclusion section.

    8. Detailed PCR data processing and PCR results such as the relative abudance of Aggregatibacter actinomycetem comitans, Porphyromonas gingivalis... should be provided.

    We thank the reviewer.  We reported the main data in Table 4. The data related to Aggregatibacter actinomycetem comitans, Porphyromonas gingivalis, etc. has not been provided since they were judged not relevant.

Reviewer 4 Report

Comments and Suggestions for Authors

To enhance the manuscript's quality, I recommend addressing these critical areas.

1. The abstract for this paper requires significant improvement. It fails to clearly outline the study's novel contributions and rationale. The methods section is overly wordy, lacking a concise summary of the overall approach and procedures. The results section is also incomplete, omitting key experimental data and findings needed to support the conclusions.

2. The abstract conclusion should provide a clear, concise summary of the study's key findings, moving beyond a simple restatement. Additionally, it should incorporate the authors' insights about the research's broader implications and significance.

3. The study measured outcomes at two time points - baseline (time 0) and after 8 weeks of toothpaste treatment. However, the authors did not provide a justification for the chosen 8-week treatment duration.

4. The Introduction should establish the context for your research and explain how it advances the knowledge in your field by building upon previous related studies. Avoid breaking the Introduction down into subsections.

5. To enable reproducibility, authors must provide comprehensive methodological details, including the sources, identities, and concentrations of all reagents and other materials used. This level of specificity is essential.

6. The discussion section lacks a clear message, fails to critically review the results, and provides an incomplete literature review. To strengthen the paper, the authors should discuss the impact of their research, outline their plans for future work, and, if applicable, offer suggestions for implementing the intervention in a specific context.

Comments on the Quality of English Language

Rephrasing certain sentences could improve the text's clarity [‘so call’, ‘p’ in the abstract] and overall coherence. To ensure a polished, fluid narrative, it is strongly recommended to engage a native English speaker or professional language editing service.

Author Response

Dear Editor,

Attached please find our reply to all of the criticisms raised by the reviewer. We feel that we did all our best to take them into consideration, to modify our text, in a way that it may better represent and discuss the main findings of our study. The following notes address in detail each specific point and give evidence of modifications in the text.

Answers Reviewer 4 Comments:

  1. The abstract for this paper requires significant improvement. It fails to clearly outline the study's novel contributions and rationale. The methods section is overly wordy, lacking a concise summary of the overall approach and procedures. The results section is also incomplete, omitting key experimental data and findings needed to support the conclusions.

We thank the reviewer. We tried to improve the text to the best of our possibilities.

  1. The abstract conclusion should provide a clear, concise summary of the study's key findings, moving beyond a simple restatement. Additionally, it should incorporate the authors' insights about the research's broader implications and significance.

We agreed with the reviewer. We changed the abstract as follows: “The toothpaste with the addition of curcumin and melatonin showed a modulation between t0 and t1 of the Campylobacter rectus (14568 vs 3532.8) and Peptostreptococcus micro (1320.8 vs 319) bacteria. In addition, modulation of pathogenic bacteria and saprophytic bacteria was shown. The synergistic action of the two additives would therefore appear to lead to promising results.”.

  1. The study measured outcomes at two time points - baseline (time 0) and after 8 weeks of toothpaste treatment. However, the authors did not provide a justification for the chosen 8-week treatment duration.

We agreed with the reviewer. In the discussion section, we suggested further studies with larger group of patients and with larger treatment period.

  1. The Introduction should establish the context for your research and explain how it advances the knowledge in your field by building upon previous related studies. Avoid breaking the Introduction down into subsections.

We agreed with the reviewer, and we have removed all the subtitles in the introduction section.

  1. To enable reproducibility, authors must provide comprehensive methodological details, including the sources, identities, and concentrations of all reagents and other materials used. This level of specificity is essential.

We thank the reviewer. We think that we have been as specific as possible. Unfortunately, we couldn’t report the formulation of the toothpaste since it is covered by patent.

  1. The discussion section lacks a clear message, fails to critically review the results, and provides an incomplete literature review. To strengthen the paper, the authors should discuss the impact of their research, outline their plans for future work, and, if applicable, offer suggestions for implementing the intervention in a specific context.

We agreed with the reviewer. We implemented the discussion section.

Rephrasing certain sentences could improve the text's clarity [‘so call’, ‘p’ in the abstract] and overall coherence. To ensure a polished, fluid narrative, it is strongly recommended to engage a native English speaker or professional language editing service.

We agreed with the reviewer. We performed an extensive revision of the article.

Round 2

Reviewer 1 Report

Comments and Suggestions for Authors

Dear authors of the article!

The updated version of the article contains changes to the materials and methods sections, results and discussion and you have done a good job.

However, some minor errors remained:

The genus Tannerella should be spelled that way, not Tannarella.

Figures 4 and 5 correct errors in microbial names in the figure caption, but not in the figures themselves.

If these minor comments are corrected, the article may be accepted for publication.

Author Response

The genus Tannerella should be spelled that way, not Tannarella.

We agreed with the reviewer. We corrected the microorganism names.

Figures 4 and 5 correct errors in microbial names in the figure caption, but not in the figures themselves.

We thank the reviewer. We checked the figures and corrected the figures themselves.

Reviewer 2 Report

Comments and Suggestions for Authors

No more comments.

Author Response

thanks to the reviewer for your work

Reviewer 3 Report

Comments and Suggestions for Authors

No issues

Author Response

thanks to the reviewer for your work